# Mammal communities are larger and more diverse in moderately developed areas

Arielle Waldstein Parsons[1,2]*, Tavis Forrester[3,4], Megan C Baker-Whatton[5], William J McShea[4], Christopher T Rota[6], Stephanie G Schuttler[1], Joshua J Millspaugh[7], Roland Kays[1,2]

[1]North Carolina Museum of Natural Sciences, Raleigh, United States; [2]Department of Forestry & Environmental Resources, North Carolina State University, Raleigh, United States; [3]Oregon Department of Fish and Wildlife, Gekeler Lane, United States; [4]Smithsonian Conservation Biology Institute, Front Royal, United State; [5]The Nature Conservancy, Fairfax Drive Arlington, Virginia; [6]Division of Forestry and Natural Resources, Wildlife and Fisheries Resources Program, West Virginia University, Morgantown, United States; [7]Wildlife Biology Program, Department of Ecosystem and Conservation Sciences, College of Forestry and Conservation, University of Montana, Missoula, United States

**Abstract** Developed areas are thought to have low species diversity, low animal abundance, few native predators, and thus low resilience and ecological function. Working with citizen scientist volunteers to survey mammals at 1427 sites across two development gradients (wild-rural-exurban-suburban-urban) and four plot types (large forests, small forest fragments, open areas and residential yards) in the eastern US, we show that developed areas actually had significantly higher or statistically similar mammalian occupancy, relative abundance, richness and diversity compared to wild areas. However, although some animals can thrive in suburbia, conservation of wild areas and preservation of green space within cities are needed to protect sensitive species and to give all species the chance to adapt and persist in the Anthropocene.
DOI: https://doi.org/10.7554/eLife.38012.001

*For correspondence:
arielle.parsons@naturalsciences.org

**Competing interests:** The authors declare that no competing interests exist.

## Introduction

Global loss of biodiversity leads to disruption of ecosystem services around the world, ultimately threatening human well-being (*Cardinale et al., 2012*). Vertebrate species loss is typically considered to be worst in the most developed landscapes, where urbanization serves as an intense and long-term disturbance that permanently alters habitat and truncates food webs (*Lombardi et al., 2017*; *McKinney, 2006*). However, for some species, urbanization can offer abundant nutrient-rich food that is less ephemeral compared to wild areas (*Bateman and Fleming, 2012*; *Wang et al., 2017*). Whether this food is enough to counteract the negative effects of disturbance (i.e. higher road mortality, fragmentation) depends on a species' ability to adapt to the stressors of urban living (*Witte et al., 1982*). Mammal species, especially those with large home ranges, are arguably most at risk from development, leading some to suggest that developed areas have a dearth of predators, and that prey species could benefit by using humans as a shield (*Crooks, 2002*; *Ordeñana et al., 2010*). Previous studies have shown cities to be depauperate of bird life, supporting the traditional view that development and biodiversity cannot coexist (*Keast, 1995*; *Strohbach et al., 2014*).

However, recent evidence has shown that some mammal species previously thought mal-adapted to urban landscapes (i.e. mountain lion [*Puma concolor*], fisher [*Martes pennanti*]) are thriving in them (*Bateman and Fleming, 2012*; *LaPoint et al., 2013*), suggesting an evolutionary trend that could be important for conservation in the Anthropocene. Existing research on mammal

**eLife digest** Humans transform natural ecosystems worldwide into towns and cities, replacing natural habitat with human-built surfaces. This loss of habitat and increase in human activity make suburban areas difficult for some species to survive in, raising concerns that developed areas become ecologically unbalanced as they lose biodiversity. However, the preservation of urban green space and lack of hunting could also open the door for some species to thrive in the midst of large human populations. Indeed, some animals, mammals in particular, have grown more tolerant of humans and appear to have adapted to suburban landscapes around the world. Some species that have been exclusively living in the wilderness, such as a small carnivore called the fisher, are even moving back into cities.

Research into how mammals are coping with the urbanization of their habitats has produced conflicting results. Studies that explore a variety of cities and habitats would help to clear up this confusion.

Parsons et al. worked with citizen scientist volunteers to survey the mammals at 1,427 sites across Washington DC and Raleigh, North Carolina. The volunteers set up motion-triggered cameras in these sites, which covered a full range of urban and wild habitats, including back yards and large nature preserves.

The cameras detected similar or higher numbers of mammal species in suburban sites compared to wild areas. Indeed, most species appear to use suburban areas at least as much as wild land. Urban green space is especially important; it is used by less urban-adapted species like coyotes to navigate areas that are densely populated by humans.

The results presented by Parsons et al. suggest that many mammals have indeed adapted to the suburban environment over the last few decades, resulting in more balanced urban ecosystems. More testing in other cities will help to determine how general this pattern of adaptation is, and provide us with knowledge that could help us to conserve many different species. However, some species were still most abundant in wild areas, emphasizing the need to also conserve wildlands and to minimize our impact on natural ecosystems.

DOI: https://doi.org/10.7554/eLife.38012.002

communities across urbanization gradients has focused on single cities, yielding conflicting results, perhaps due to variation in city structure and characteristics (*Lombardi et al., 2017*; *Saito and Koike, 2013*). Given the rapid expansion of urban areas worldwide, and the recent case studies of urban adaptations by wildlife (*LaPoint et al., 2013*; *Riley et al., 2014*; *Wang et al., 2017*), more large-scale studies are needed to evaluate the response of wildlife communities to urban development if we are to understand urban ecology, conservation, and evolution in the Anthropocene.

Here, we present the results of a large-scale mammal survey of two urban-wild gradients. Our objectives were to determine how diversity, richness, detection rate, and occupancy of the mammal community change as a function of human disturbance. We hypothesized that the availability of supplemental food at higher levels of development would positively affect mammalian populations and outweigh the negative effects of disturbance, except for the most sensitive species. Specifically, we predicted that mammalian relative abundance would increase with developmental level but that species richness and diversity would decrease. Furthermore, we predicted that occupancy of the most sensitive species (i.e. large and medium carnivores) would be highest in wild areas both in our study area and around the world.

## Materials and methods

### Study sites

Washington, District of Columbia, USA (hereafter DC) is a city of approximately 177 km$^2$ with an estimated human population size of 681,000, thus a density of 3847 people/km$^2$. Our study spanned a 56,023.7 km$^2$ area around the city with a mean of 4.4 houses/km$^2$ and matrix of agriculture (~21.3%) and forest (~54.1%). Raleigh, North Carolina, USA (hereafter Raleigh) is approximately 375 km$^2$ with an estimated human population size of 459,000, thus a density of 1278 people/km$^2$. Our study

spanned a 66,640 km$^2$ area around the city with a mean of 17.7 houses/km$^2$ and matrix of agriculture (~24.3%) and forest (~52.3%).

## Citizen science camera trap surveys

From 2012–2016, 557 trained volunteers deployed 1427 unbaited camera traps across an urban-wild gradient around Raleigh and DC. Each individual camera was considered a 'camera site' and volunteers ran cameras at an average of two sites each. Following Hammer et al. (2004), we used the Silvis housing density dataset with 1km grid cells to define five development levels of the gradient for sampling stratification (excluding open water): urban (>1000 houses/km$^2$), suburban (147.048–1000 houses/km$^2$), exurban (12.64–147.047 houses/km$^2$), rural (0.51–12.63 houses/km$^2$) and wild (<0.5 houses/km$^2$). Within those gradient levels, camera placement was also stratified between residential yards, open areas (>0.001 km$^2$ absent of trees), small forest fragments ($\leq$ 21 km$^2$) and large forest fragments (>1 km$^2$) *Supplementary file 1*. Forest fragment size was verified using the 2006 US National Landcover Dataset (NLCD) and Landscape Fragmentation Tool v2.0 (Vogt et al., 2007) in ArcMap (Version 10.1, ESRI, Redlands, CA, USA) which defines forest fragments by size. Yards were not available for sampling in the urban or wild levels of the gradient. Urban areas were not sampled in Raleigh and open areas were not sampled in DC. All adjacent cameras were spaced at least 200 m apart. Camera placement was randomized as much as possible using ArcMap (Version 10.1) to randomly generate points within polygons while following certain rules. For example, we selected sites within forests that volunteers were permitted to access and were within a reasonable hiking distance (i.e. < 11 km hike round trip) with terrain that was not too steep to traverse safely (i.e. <45 degree slope). Within yards, cameras were placed as randomly as possible while avoiding the highest human traffic areas (i.e. walkways, doors, gates and driveways).

No explicit power analysis was used to predetermine sample size. Our sample size goal was 20 spatial replicates (equating to ~420 trap nights), which has been found to maximize precision for estimating detection rate (*Kays et al., 2010*; *Rowcliffe et al., 2008*). Camera sites are biological replicates, parallel measurements capturing random biological variation. This study did not include technical replicates.

Volunteers used Reconyx (RC55, PC800, and PC900, Reconyx, Inc. Holmen, WI) and Bushnell (Trophy Cam HD, Bushnell Outdoor Products, Overland Park, KS) camera traps attached to trees at 40 cm above the ground. Cameras were deployed for three weeks and then moved to a new location without returning, with sampling taking place continuously throughout the year. Cameras recorded multiple photographs per trigger, at a rate of 1 frame/s, re-triggering immediately if the animal was still in view. We grouped consecutive photos into on sequence if they were <60 s apart, and used these sequences as independent records, counting animals in the sequence, not individual photos (*Parsons et al., 2016*). We then collapsed these independent records into daily detection/non-detection for occupancy modeling. Initial species identifications were made by volunteers using customized software (available freely from eMammal.org, source code proprietary) and all were subsequently reviewed for accuracy before being archived at the Smithsonian Digital Repository (*McShea et al., 2016*).

## Diversity

We used package iNEXT (*Hsieh et al., 2016*) in R (Version 3.1.0; *R Development Core Team., 2008*) via R Studio (*RStudio Team, 2015*) to calculate Hill numbers (i.e. the effective number of species, incorporating relative abundance and richness) of species richness and Shannon diversity (*Chao et al., 2014*) between gradient levels (urban-suburban-exurban-rural-wild) and plot types (yard, open, small forest, large forest). iNEXT calculates the Shannon diversity as Hill number q = 1, equal to the exponential of Shannon's entropy index, thus the natural log of those results was used for display purposes. We used detection/non-detection data to compute diversity estimates and the associated 95% confidence intervals via rarefaction, plotting the diversity estimates while accounting for sample size. We fit a curve to diversity estimates between gradient levels using a generalized additive model with a polynomial term.

## Model covariates

We modeled variation in occupancy (ψ) and detection rate using 13 covariates (*Supplementary file 2*) representing development level, the amount of core forest, small scale forest cover, prey relative abundance and whether hunting was allowed. We added year as a covariate to account for population changes over time. We used the Landscape Fragmentation Tool v2.0 (*Vogt et al., 2007*) and the NLCD (2006) land use dataset in ArcMap (Version 10.1) to create a landcover layer representing the percent of large core forest (forest patches larger than 1 km$^2$) in a 5 km radius around camera locations which we considered best approximated the home range size of our target species (*Bekoff, 1977*; *Fritzell and Haroldson, 1982*; *Lariviere and Pasitschniak-Arts, 1996*; *Lariviere and Walton, 1997*). Forest patches did not necessarily fall entirely within the buffer. We considered road density as an additional covariate at the 5 km scale but initial evaluations showed it to be highly correlated with housing density (87.1%) so we chose to eliminate it from the analysis. We used a 100 m radius for small-scale forest cover to best represent small forest patches within suburban neighborhoods (e.g. small vacant lots with trees, greenways). We represented deer and rodent+lagomorph relative abundance using site-specific detection rate (the number of detections divided by the total number of camera-nights). We included an indicator (0/1, no hunting/hunting) to categorize whether a site allowed hunting or not. We modeled detection probability (p) using five covariates (*Supplementary file 2*). Because both ambient temperature and undergrowth can affect the camera's ability to detect an animal, we included daily covariates for temperature and NDVI (Moderate Resolution Imaging Land Terra Vegetation Indices 1 km monthly, an average value over the month(s) the camera ran) obtained from Env-DATA (*Dodge et al., 2013*). To complement NDVI, we also considered site-specific detection distance, a measure of how far away the camera was able to detect a human, which is influenced by both understory and site topography. We included an indicator (0/1, not yard/yard) to categorize whether a site was a residential yard or not. In Raleigh, two different camera models were used (both Reconyx and Bushnell) so we added a 0/1 (Bushnell/Reconyx) covariate to account for potential difference in detection probability between the two brands. We diagnosed univariate correlations between covariates using a Pearson correlation matrix, and used a restrictive prior for beta coefficients where correlation was >0.60 (i.e. logistic(0,1); a prior with reduced variance to induce shrinkage, similar to ridge regression; *Hooten and Hobbs, 2015*). All covariates were mean-centered.

## Detection rate models

We used a Poisson count model (e.g. *Kays et al., 2017*) to assess differences in total mammal detection rate (i.e. the intensity with which a site was used, count/day) between the five gradient levels (urban, suburban, exurban, rural, wild) and four plot types (large forest, small forest, open, yard). We fit a curve to total detection rate estimates between gradient levels using a generalized additive model. No other covariates were used in this model. We then ran separate count models for four predator species (coyote (*Canis latrans*), gray fox (*Urocyon cinereoargenteus*), red fox (*Vulpes vulpes*) and bobcat [*Lynx rufus*]) to evaluate covariates of detection rate, running one fully-parameterized model (*Supplementary file 2*) to evaluate which explained the most variation in detection rate. We assessed model fit with posterior predictive checks (PPC) (*Gelman et al., 2014*; *Kery and Schaub, 2012*) by calculating the sum of squared Pearson residuals from observed data ($T(y)$) and from data simulated assuming the fully parameterized model was the data-generating model ($T(y_{sim})$). We calculated a Bayesian p-value as $p_B = \Pr(T(y_{sim}) > T(y))$ from posterior simulations and assumed adequate fit if $0.1 < p_B < 0.9$ (*Supplementary file 3*). We fit the detection rate model in OpenBUGS v3.2.3 (*Lunn et al., 2009*) via R2OpenBUGS v3.2 (*Sturtz et al., 2005*) in R (Version 3.1.0) via R Studio. We based inference on posterior samples generated from three Markov chains, using trace plots to determine an adequate burn-in phase. All models achieved adequate convergence ($R \leq 1.1$) (*Gelman et al., 2014*) by running for 50,000 iterations following a burn-in phase of 1000 iterations, thinning every 10 iterations. We based significance on whether parameter 95% credible intervals overlapped zero.

## Occupancy models

We used the multispecies occupancy model of *Rota et al. (2016)* to estimate the probability of occupancy of four predator species: bobcat, coyote, red fox and gray fox. Although we are using

the term occupancy, because data were collected from camera traps estimates are more analogous to 'use' than true occupancy (*Burton et al., 2015*). This model is distinct from the classic multispecies community models (*Dorazio and Royle, 2005*;*Dorazio et al., 2006*; *Gelfand et al., 2005*) and is rather a generalization of the single-season occupancy model (*MacKenzie et al., 2002*) to accommodate two or more interacting species. It contains single-species (first order) occupancy models for each interacting species alone as well as pairwise (second order) models for the co-occurrence of each pair of species (*Rota et al., 2016*). For each species and pairwise interaction, the model estimates detection probability ($p$), defined as the probability of detecting an occurring species at a camera site, and occupancy ($\psi$), defined as the probability that a given camera site is occupied, for each species. The latent occupancy state of each species at a site is modeled as a multivariate Bernoulli random variable such that (assuming 2 interacting species):

$$Z \sim MVB(\psi_{11}, \ \psi_{10}, \ \psi_{01}, \ \psi_{00})$$

Where $\psi_{11}$ is the probability that both species occupy a site, $\psi_{10}$ is the probability that only species 1 occupies a site, $\psi_{01}$ is the probability that only species 2 occupies a site and $\psi_{00}$ is the probability that neither species occupies a site. We assumed all species occurred independently and considered the same set of five covariates for the detection probability models and 13 covariates in the occupancy model of each species (*Supplementary file 2*). We considered interactions (i.e. city*covariate) between each occupancy covariate and city (0/1, DC/Raleigh). We estimated occupancy for each species across levels of the development gradient (urban, suburban, exurban, rural, wild) and plot types (yard, open, small forest, large forest) within each city separately by including development level and plot type as categorical covariates in our model.

We fit models in STAN (Version 2.15.1; *Stan Development Team, 2015b*) via the RSTAN (Version 2.15.1; *Stan Development Team, 2015a*) interface in R (Version 3.4.0) via R Studio (Version 1.0.143). We based inference on posterior samples generated from two Markov chains, using trace plots to determine an adequate burn-in phase and subsequently running chains until they reached adequate convergence ($R > 1.1$) (*Gelman et al., 2014*). All models achieved adequate convergence by running for 3000 iterations following a burn-in phase of 1000 iterations. We based predictor significance on whether beta coefficient 95% credible intervals overlapped zero. We assessed model fit with posterior predictive checks (PPC) (*Gelman et al., 2014*; *Kery and Schaub, 2012*) by calculating the sum of squared Pearson residuals from observed data ($T(y)$) and from data simulated assuming the fully parameterized model was the data-generating model ($T(y_{sim})$). We calculated a Bayesian $p$-value as $p_B = \Pr(T(y_{sim}) > T(y))$ from posterior simulations and assumed adequate fit if $0.1 < p_B < 0.9$. To our knowledge, the squared Pearson's residual has not been derived in the context of occupancy models, so we present our derivation of this test statistic in *Supplementary file 4*. We added a random effect on detection/non-detection for the coyote portion of the model since initial assessments of fit for this species were inadequate (i.e. $p_B > 0.9$). We assessed differences in occupancy between gradient levels for each species using overlapping 95% confidence intervals.

## Comparison with global occupancy data

We removed omnivores from the dataset of *Rich et al. (2017)* to better compare with carnivore occupancy from our own dataset. Where species occupancy was estimated from multiple studies in the Rich et al. dataset, we calculated averages to compare to occupancy estimates from our own study. We summarized occupancy estimates of Rich et al. and our own study within each developmental level using a box and whisker plot and assessed statistically significant differences based on whether or not interquartile ranges overlapped.

## Data accessibility

Raw detections data have been deposited in Data Dryad, doi:10.5061/dryad.11rf64v. The software used for initial species identifications is available via eMammal.org. To download and use the software, users must first create an account on eMammal and become associated with an existing project. This can be done by using the 'Join' button on the project's homepage, or by emailing the contact person, also listed on the project homepage. Usually the user will also have to pass an online or in person training, depending on the project requirements, to be approved to download the software.

## Results and discussion

Working with citizen scientist volunteers, we obtained 53,273 detections of 19 mammal species at 1427 sites along an urban-wild gradient in Washington, DC and Raleigh, NC, USA, sampling both private and public lands. In DC, we detected 17 mammal species with mean naïve occupancy of 0.19 (min = 0, max = 0.93) and mean detection rate of 0.09 detections/day (min = 0, max = 1.05). In Raleigh, we detected 17 mammal species with mean naïve occupancy of 0.14 (min = 0, max = 0.79) and mean detection rate of 0.08 detections/day (min = 0, max = 0.09).

We found no significant decline of species diversity or richness from suburban to wild gradient levels (*Figure 2—figure supplement 1*, *Figure 1*). However, Shannon diversity was significantly lower at the urban level in DC, possibly due to low sampling (*Figure 2*, *Supplementary file 1*). Diversity in yards was significantly higher or not statistically different from large and small forest fragments in both cities (*Figure 2—figure supplements 2,3*). Most (92.3%) of the 13 mammal species

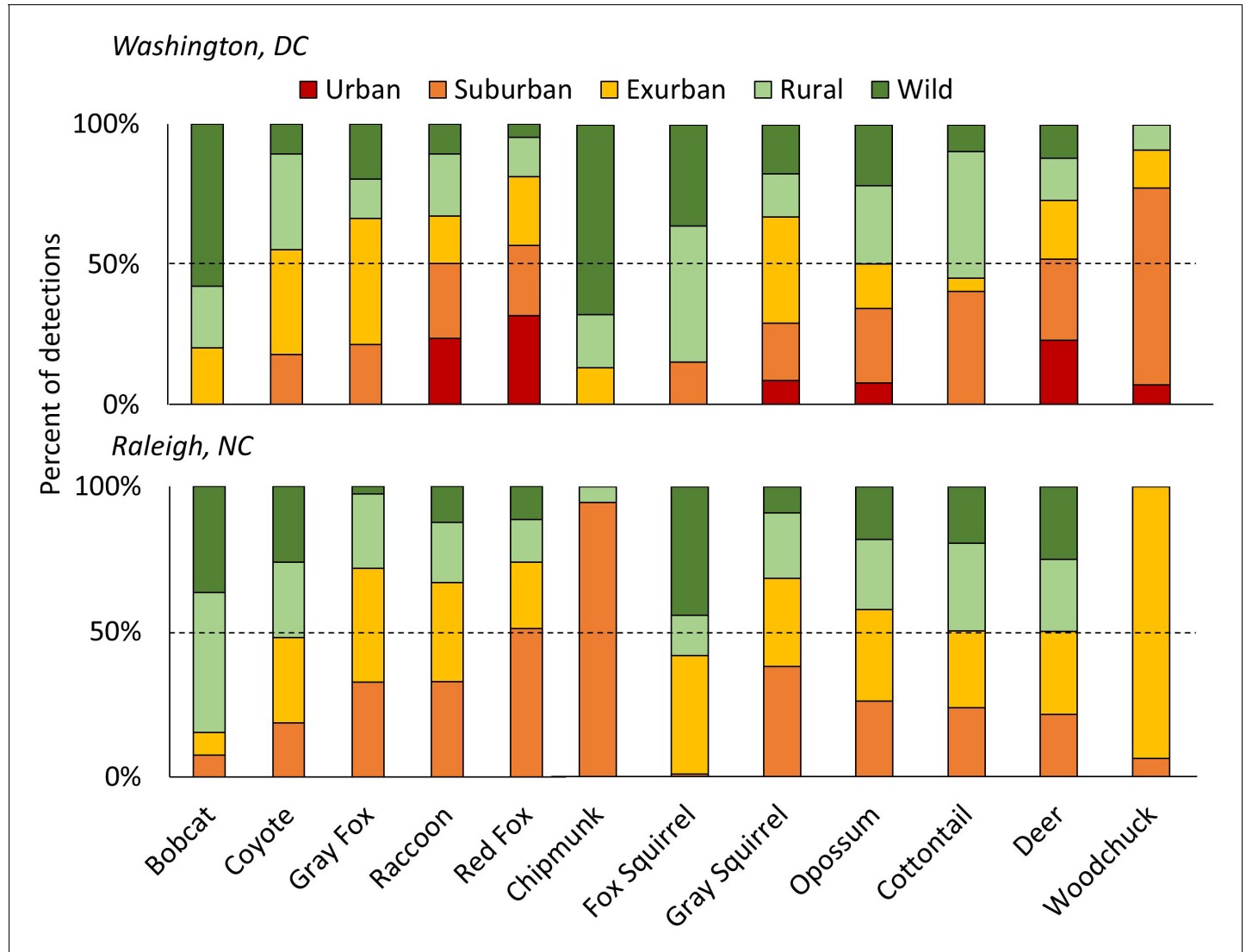

**Figure 1.** The percent of detections for each species of carnivores (left) and herbivores (right) in each development level along the urban-wild gradient in Washington, DC and Raleigh, NC, USA accounting for the effort (i.e. camera nights) within each level, sorted from lowest to highest proportion urban/suburban in DC. The dashed line shows 50% of total detections. Some species were predominantly rural/wild (i.e. bobcats and fox squirrels) while others were mainly detected in urban/suburban habitats (i.e. red fox, raccoon). Patchy distributions at different gradient levels were seen for species at the edge of their ranges (i.e. chipmunks and woodchucks in Raleigh). Urban habitats were not sampled in Raleigh.
DOI: https://doi.org/10.7554/eLife.38012.003

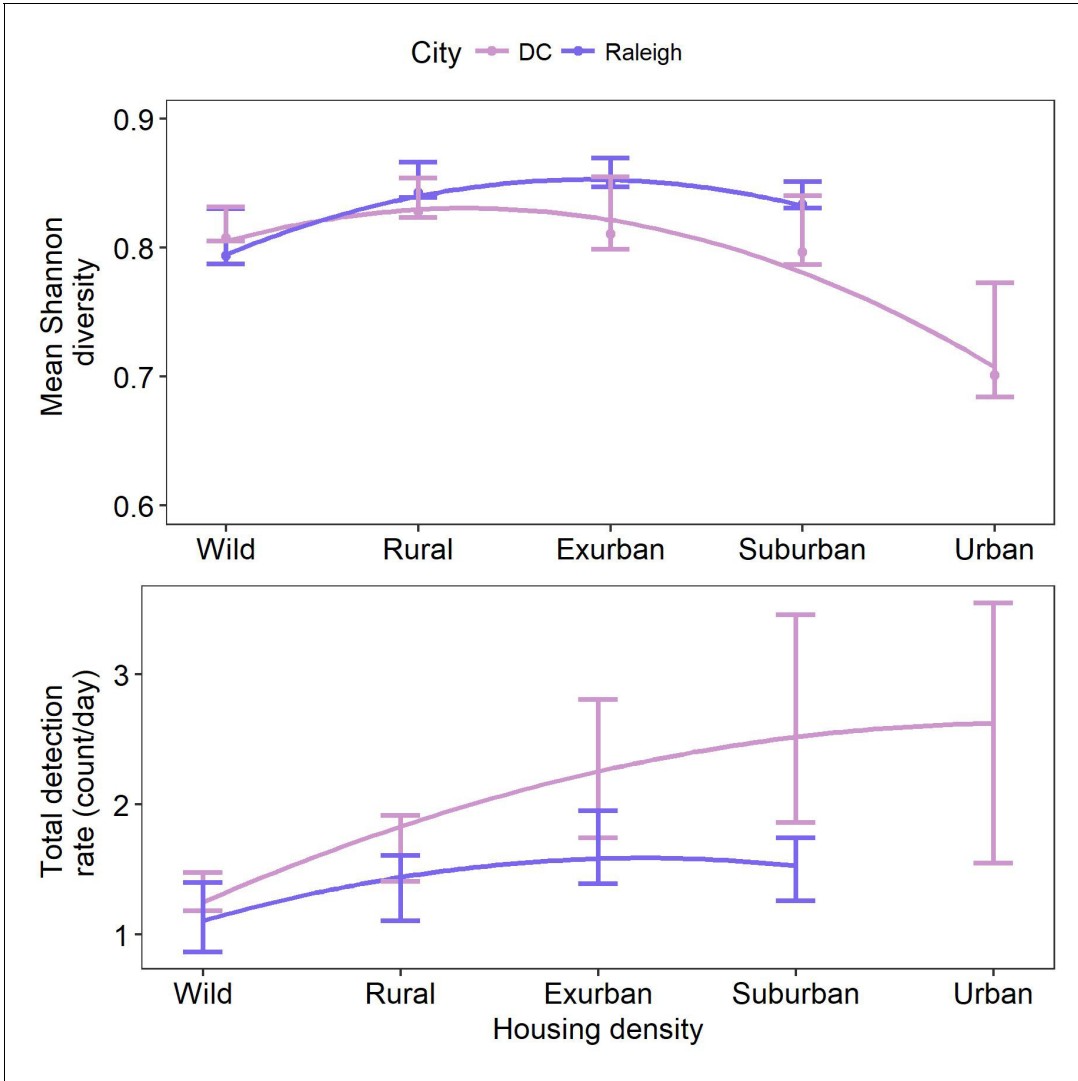

**Figure 2.** Mean Shannon diversity and total detection rate along a gradient of housing density in two cities, Washington, DC and Raleigh, NC USA taken from camera traps. Bars show 95% confidence intervals, lines are fit using a generalized additive model with a polynomial term. Diversity peaked at intermediate levels of urbanization (exurban in DC and suburban in Raleigh). Total detection rate peaked at the urban level in DC and exurban level in Raleigh.

DOI: https://doi.org/10.7554/eLife.38012.004

The following figure supplements are available for figure 2:

**Figure supplement 1.** Rarefaction curves estimating species richness in five development levels (urban, suburban, exurban, rural, wild) in two cities, Washington, DC and Raleigh, NC, USA, using camera traps between 2012 and 2016.

DOI: https://doi.org/10.7554/eLife.38012.005

**Figure supplement 2.** Shannon diversity index estimates from camera trapping in two cities, Washington, DC and Raleigh, NC, USA, across five development levels (urban, suburban, exurban, rural, wild).

DOI: https://doi.org/10.7554/eLife.38012.006

**Figure supplement 3.** Rarefaction curve estimating species richness in three plot types (residential yard, small forest, large forest) in two cities, Washington, DC and Raleigh, NC, USA, using camera traps between 2012 and 2016.

DOI: https://doi.org/10.7554/eLife.38012.007

detected >20 times occupied all levels of development below the urban level. Two of the largest predators, coyotes and bobcats, were absent from the highest development level (urban) but were detected at all other levels in both cities. Black bears (*Ursus americanus*), which are actively discouraged from colonizing central North Carolina (*North Carolina Wildlife Resources Commission, 2011*), were not detected in Raleigh and were detected in DC at all levels of the gradient except

suburban and urban, though were predominantly in the wild level. These results indicate that the extant mammal guild exploits all levels of the urban-wild gradient and that no species are entirely relegated to the wild gradient level. However, some species appear less adapted to habitation in human-dominated areas, spending most of their time at the wild levels of the gradient (i.e. bobcat, bear; *Figure 1*). We recognize that the current community represents species that survived the initial arrival of high-density human settlement. In particular, two large predators (wolves (*Canis lupus*) and cougars [*Puma concolor*]) were extirpated from our study area a century ago. However, even cougars and wolves have recently shown surprising adaptability in the face of development at other sites (*Bateman and Fleming, 2012*; *Wang et al., 2017*) suggesting that, given enough time and protection from persecution, many of the most 'wild' of species may adapt to human development.

Predators are thought to be the most at risk from urbanization (*Crooks, 2002*), therefore, we evaluated predictors for occupancy (*MacKenzie et al., 2002*) and detection rate (*Kays et al., 2017*) for four carnivores: coyote, gray fox, red fox, and bobcat. Both of our models fit well, with Bayesian p-values between 0.1 and 0.9 (*Supplementary file 3*). Suburban and urban occupancy probabilities were not statistically different from wild for any of the species (*Figure 3—figure supplement 1*) and we noted a decreasing trend in occupancy from urban to wild (*Figure 3*). We compared the occupancy estimates from our study to those reported for carnivores in protected areas around the world (*Rich et al., 2017*) and found no significant difference (*Figure 3*), suggesting that the ecological function of predators in this urban system is not substantially reduced from the current wild state, excepting the historical extirpation of the two largest native predators from the region.

Our occupancy and detection rate models yielded similar results (*Supplementary file 5–7*) demonstrating that green space is important to carnivore species that are less-adapted to human-altered landscapes. These models show a greater association of carnivores with green space when housing density is high (e.g. coyote and gray fox, *Supplementary file 6*, *7*), consistent with other studies finding urban green space important in maintaining biodiversity in cities (*Gallo et al., 2017*; *Lombardi et al., 2017*; *Matthies et al., 2017*). It is likely that shyer species are not avoiding regions of high human density, but require patches of forest to navigate residential areas that are freely used by more commensal species, such as red foxes (*Tigas et al., 2002*), which we frequently detected in yards. Indeed, we found a gradient of responses in carnivore use of human-dominated environments, from red fox which are the most urban adapted (i.e. negatively associated with local large forest fragments and the only species to have a positive association with yards) to bobcats which appear to be the most human-averse (i.e. rarely detected in the suburban level of the gradient) (*Figure 1*; *Figure 3—figure supplement 1*).

Contrary to expectations, we found no evidence for a negative impact of suburban and exurban development on extant native mammal diversity, richness, and occupancy and detection rate of carnivores. In fact, all metrics were significantly greater than, or equal to, wild areas. We suspect that developed areas offer good food resources for wildlife through direct and indirect feeding (i.e. bird feeders supplementing prey, pets), accidental feeding (i.e. garbage), and ornamental plantings (for herbivores), but testing this hypothesis will require additional diet studies in urban landscapes (*Contesse et al., 2004*). Furthermore, the structure of mature suburbia (i.e. older, established neighborhoods with large trees, wooded riparian areas, small parks) contributes to a more diverse and varied landscape than wild areas with more homogenous forest cover, which is potentially beneficial for many generalist species. Developed areas where hunting is limited or prohibited also offer a safe haven for game species, presuming they can navigate the road networks (*Collins and Kays, 2011*) and avoid direct human conflict.

Our discovery of a wild suburbia suggests high levels of adaptation by mammals to developed landscapes over the last few decades, including predators and prey. The resilience of these species gives hope for wildlife in the Anthropocene, but the generality of this pattern needs to be tested in other cities to understand how habitat type, development patterns, apex predators, and hunting regulations influence urban mammal communities, as there are examples of far more drastic impacts of urbanization on other taxa and in other places around the globe (*Keast, 1995*; *McKinney, 2008*). Indeed, in Tokyo, Japan, the relative abundance of mammals declined with urbanization (*Saito and Koike, 2013*) and avian communities in Quebec, Canada and Rennes, France showed a similar decline in richness (*Clergeau et al., 1998*; *Saito and Koike, 2013*). This suggests that city structure, size and human density may influence mammalian distribution along urban-wild gradients with large, sprawling New World cities showing different patterns than the smaller more concentrated cities of

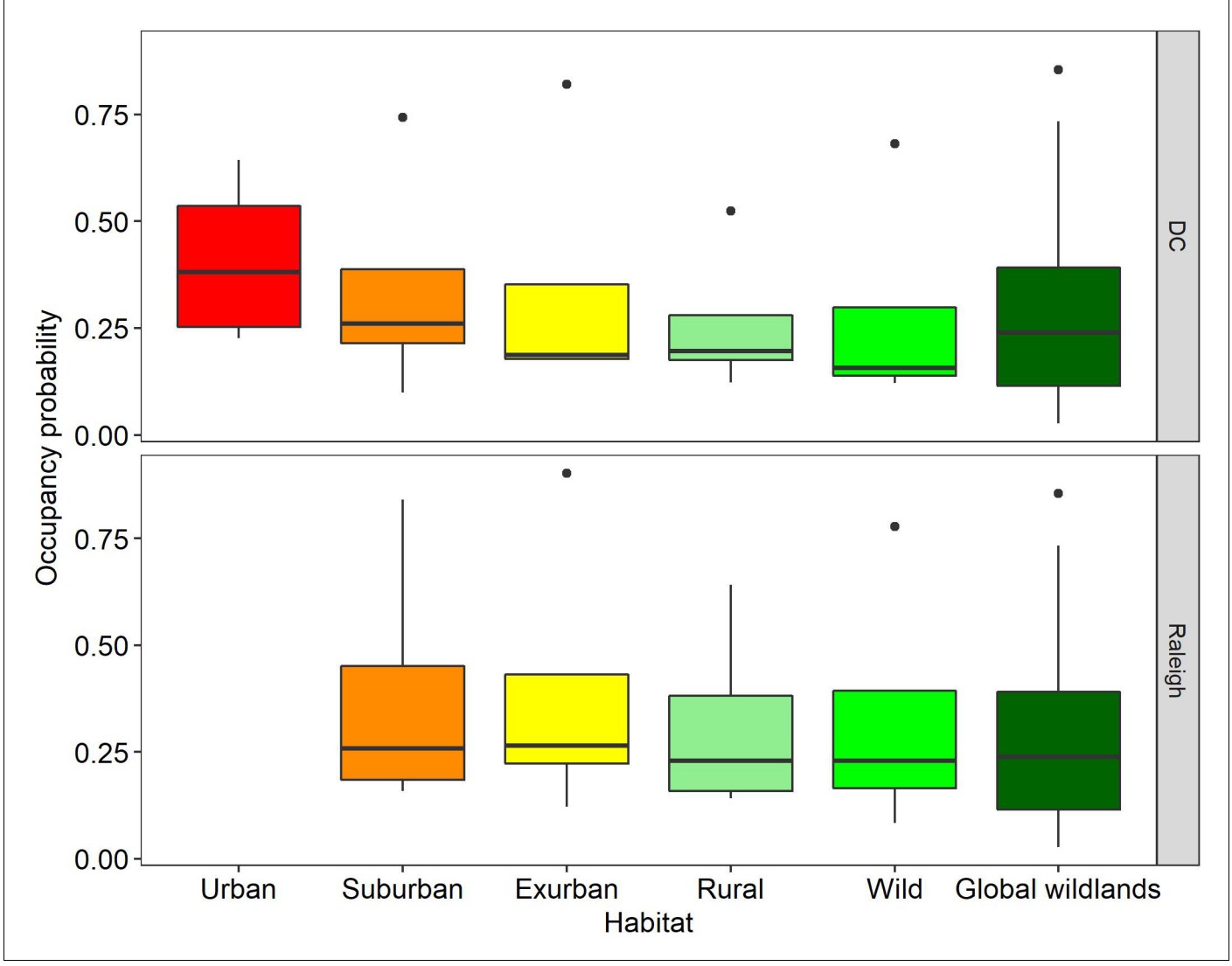

**Figure 3.** Comparison of carnivore (i.e.bobcat, coyote, gray fox, red fox) occupancy probabilities at each developmental level in two cities with global values from Rich et al. (*Rich et al., 2017*), representing 93 carnivores from 13 protected areas on five continents (Global Wildlands). Each box for our dataset represents the distribution of marginal occupancy probabilities for each of four carnivore species in that city (i.e. four probabilities). The boxes for Global Wildlands represent the distribution of marginal occupancy probabilities for 93 species. We found no statistically significant differences between any habitat levels in our study or between our study and global wildland occupancy probabilities but noted a decreasing trend in occupancy from urban-wild. We included only predators from *Rich et al. (2017)* and removed omnivores (i.e. raccoon, coati) to better reflect our data.
DOI: https://doi.org/10.7554/eLife.38012.008

The following figure supplement is available for figure 3:

**Figure supplement 1.** Occupancy estimates from single season occupancy model for four carnivore species (bobcat, coyote, gray fox and red fox) in five development levels (urban, suburban, exurban, rural, wild) in two cities, Washington, DC and Raleigh, NC, USA, using camera traps between 2012 and 2016.
DOI: https://doi.org/10.7554/eLife.38012.009

the Old World. Although our study provides a less dire picture of urban ecosystem function than previously thought, we do not suggest abandoning mitigation of urbanization's negative impacts, or conservation of completely wild areas. Factors such as urban green space, connectivity and availability of completely wild areas give species the time and space to adapt to changing habitats and climates. Further understanding of how urban wildlife navigates human-dominated areas and factors that contribute to the adaptation of species to the Anthropocene will be critical to maintaining diversity in a rapidly urbanizing world.

## Acknowledgements

We thank our 557 volunteers for their hard work collecting camera trap data for this study. For their field assistance and volunteer coordination we thank the staff of the NPS, USFWS, USFS, TNC, NC, VA, and MD State Parks, NCWRC and VDGIF. We thank A Mash, N Fuentes, S Higdon, T Perkins, L Gatens, R Owens, R Gayle, C Backman, K Clark, J Grimes and J Simkins for their help reviewing photographs. This work was conducted with funding from the National Science Foundation [grant #1232442 and #1319293], the USDA National Institute of Food and Agriculture, the VWR Foundation, the US Forest Service, the North Carolina Museum of Natural Sciences and the Smithsonian Institution. We thank M Katti and R Dunn for comments on early versions of this manuscript.

## Additional information

### Funding

| Funder | Grant reference number | Author |
| --- | --- | --- |
| National Science Foundation | #1232442 | William J McShea<br>Roland Kays |
| National Science Foundation | #1319293 | William J McShea<br>Roland Kays |
| VWR Foundation | | William J McShea<br>Roland Kays |
| International Programs, US Forest Service | | William J McShea<br>Roland Kays |
| Smithsonian Institution | | William J McShea |
| National Institute of Food and Agriculture | McIntire Stennis project , WVA00124 | Christopher T Rota |
| North Carolina Museum of Natural Sciences | | Roland Kays |

The funders had no role in study design, data collection and interpretation, or the decision to submit the work for publication.

### Author contributions

Arielle Waldstein Parsons, Conceptualization, Data curation, Formal analysis, Validation, Investigation, Visualization, Methodology, Writing—original draft, Project administration, Writing—review and editing; Tavis Forrester, Conceptualization, Supervision, Funding acquisition, Investigation, Methodology, Project administration, Writing—review and editing; Megan C Baker-Whatton, Conceptualization, Supervision, Investigation, Methodology, Project administration, Writing—review and editing; William J McShea, Conceptualization, Data curation, Supervision, Funding acquisition, Investigation, Methodology, Project administration, Writing—review and editing; Christopher T Rota, Formal analysis, Visualization, Writing—review and editing; Stephanie G Schuttler, Investigation, Writing—review and editing; Joshua J Millspaugh, Formal analysis, Writing—review and editing; Roland Kays, Conceptualization, Supervision, Funding acquisition, Validation, Investigation, Methodology, Writing—review and editing

### Author ORCIDs

Arielle Waldstein Parsons (iD) http://orcid.org/0000-0003-1076-2896

### Decision letter and Author response

Decision letter https://doi.org/10.7554/eLife.38012.021
Author response https://doi.org/10.7554/eLife.38012.022

## Additional files

### Supplementary files

• Supplementary file 1. Effort expressed as camera nights with spatial replicates in parentheses for camera traps run in Washington, DC and Raleigh, NC from 2012 to 2016 between different levels along the urban-wild gradient around each city.
DOI: https://doi.org/10.7554/eLife.38012.010

• Supplementary file 2. Covariates used in the detection rate and occupancy analyses.
DOI: https://doi.org/10.7554/eLife.38012.011

• Supplementary file 3. Results of goodness-of-fit tests for occupancy and Poisson count models assessed by posterior predictive check with adequate fit if $0.1 < pB < 0.9$.
DOI: https://doi.org/10.7554/eLife.38012.012

• Supplementary file 4. Calculating goodness of fit statistics
DOI: https://doi.org/10.7554/eLife.38012.013

• Supplementary file 5. Results of a Poisson regression to determine differences in detection rate between the wild gradient level (reference level) and all other levels of the development gradient (above dotted line). Also presented are results of a separate Poisson regression to determine differences in detection rate between yards and all other plot types. Significant results (95% CIs not overlapping zero) are in bold.
DOI: https://doi.org/10.7554/eLife.38012.014

• Supplementary file 6. Results of Poisson regression for single species using camera trapping over two cities between 2012–2016. Posterior mean and posterior standard deviation for each predictor are shown with bold entries indicating predictors with 95% credible intervals that did not overlap zero. Predictors in bold were used for modeling occupancy for that species. Housing density was used as a predictor in all occupancy models, regardless of whether it was significant in the preliminary count analysis.
DOI: https://doi.org/10.7554/eLife.38012.015

• Supplementary file 7. Beta coefficients and 95% credible intervals (parentheses) for an occupancy model based on camera trapping data in Washington, DC, USA and Raleigh, NC, USA from 2012 to 2016. Those coefficients with 95% CIs not overlapping zero are shown in bold.
DOI: https://doi.org/10.7554/eLife.38012.016

• Transparent reporting form
DOI: https://doi.org/10.7554/eLife.38012.017

### Data availability

Raw detections data have been deposited in Data Dryad, doi:10.5061/dryad.11rf64v. The software used for initial species identification is available via eMammal.org. To download and use the software, users must first create an account on eMammal and become associated with an existing project. This can be done by using the 'Join' button on the project's homepage, or by emailing the contact person, also listed on the project homepage. Usually the user will also have to pass an online or in person training, depending on the project requirements, and they will then become approved to download the software.

The following dataset was generated:

| Author(s) | Year | Dataset title | Dataset URL | Database, license, and accessibility information |
|---|---|---|---|---|
| Arielle Waldstein Parsons, Tavis Forrester, Megan C Baker-Whatton, William McShea, Christopher T Rota, Stephanie G Schuttler, Joshua J Millspaugh, Roland Kays | 2018 | Wild suburbia: mammal communities are larger and more diverse in moderately developed areas | http://dx.doi.org/10.5061/dryad.11rf64v | Available at Dryad Digital Repository under a CC0 Public Domain Dedication |

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
