## [Decision Letter]

Thank you for submitting your article "Wild suburbia: mammal communities are larger and more diverse in moderately developed areas" for consideration by *eLife*. Your article has been reviewed by four peer reviewers, including Bernhard Schmid as the Reviewing Editor and Reviewer #1, and the evaluation has been overseen by Ian Baldwin as the Senior Editor. The following individuals involved in review of your submission have agreed to reveal their identity: Marc Kéry (Reviewer #3) and Ingolf Kühn (Reviewer #4).

The reviewers have discussed the reviews with one another and the Reviewing Editor has drafted this decision to help you prepare a revised submission.

Summary:

The value of this paper is that it addresses a question of broad relevance with novel methods. The question is whether human population density near two US cities reduces the occurrence of large mammals in four types of habitats, large forests, forest fragments, open areas and yards. The assessment method involved 557 citizens operating cameras deployed for around 20 days at each of 1427 locations throughout a period of 4 years. In addition, the authors compared the results of this assessment with a global data set.

The result was that mammal occurrence did not decline with human population density and even showed an opposite tendency, except for bobcats. Even though this results is not "strong" in the sense that the main point is the absence of a negative influence of human population density, it does send a very important message, in particular because it is based on such a large sampling scheme with a clear design of five human population density levels (wild < rural < exurban < suburban < urban) factorially crossed with the four habitat types (obviously, these two factors were not fully orthogonal because for example open areas and yards could not be found in the wild).

Essential revisions:

The interpretation of human population density as equivalent to disturbance is less important than implied by the authors and should be better justified. In particular, the comparison with the so-called intermediate disturbance hypothesis at the start of the Results and Discussion section seems unnecessary and unjustified, because it is not really tested later on nor even discussed. In fact, the curves drawn in Figure 1 are not really hump-shaped and probably do not have significant downwards curvature at high human population density (interpreted as high disturbance). The hump is even more elusive if Figure 3 is inspected.

Currently, the discussion is somewhat US-centered. Although the two case-study cities are from that region, it would make the paper more relevant if comparisons with literature from other regions, including Europe, and involving other taxonomic groups would be included. In this context it would also be important to put the levels of human population densities in the two studied cities in relation to cities in the old world, which often occupy much smaller areas.

The major issue you need to address in the revision is the description of methods. The detailed comments from the reviewers in this regard are listed below:

I am not sure whether the results are influenced by a measurement bias, i.e. species tending to have higher activity in more disturbed regions and hence being more frequently recorded.

Subsection “Detection rate models”: Either refer from here to the last paragraph of the “Occupancy models” subsection or mention already here how large (long) the burn-in phase was and report the value of i.

Subsection “Occupancy models”: To me it is not clear which species were paired: all ones, or just predators?

Subsection “Occupancy models”: Explain the elements of the formula (including subscripts).

Subsection “Comparison with global occupancy data”: Be explicit on how this was assessed.

Currently, the information about the study design is placed in different parts of the manuscript so that it is difficult to get the overview. Supplementary file 1 is the most useful in this context, even though it would be easier if number of camera locations would also be shown (in addition to number of nights). It is not clear, if there was an additional spatial stratification that was not used in the analyses. For example, I could imagine that within each cell in Supplementary file 1 there were some spatial units and within these spatial units camera locations. Even if that was not the case, the question remains if spatial distance and arrangement of camera locations should have been considered.

Why did you use the crude models of rarefaction rather than obtain the inferences on species richness (including species accumulation) from the multi-species occupancy model of Rota et al.? See Dorazio et al. (2006) for how this can be done. Also, please cite the important work of Dorazio and Royle (2005) and Dorazio et al. (2006) as a foundation on which models such as that by Rota et al. are built.

- Subsection “Model covariates” the sentence “We represented whether a site allowed hunting or not using 0/1.” sounds strange, rewrite.

- Subsection “Model covariates”: Please explain what a restrictive prior is.

- Subsection “Occupancy models”: Really, the other basic groundstone on which your work is based are the multispecies occupancy models of Dorazio and Royle and of Gelfand (independently developed in 2005). These should also be cited.

- Subsection “Occupancy models”: “We assumed all species occurred independently…”. Does this fit with what was just said “It contains single-species (first order)…”?

Subsection “Data reporting”: The first part of this paragraph is a bit weird to me. If you have to have a formal paragraph like this for the journal specifications, please ignore. As of now it reads like you are trying to convince the reader why an observational study was chosen instead of a true experiment. I think it is given that a camera trap study is going to be more of an observational study – we just try to standardize our sampling design as much as possible. If this paragraph is not required, I would just go straight into your methods – i.e. study design, site selection, sample size, etc.

Subsection “Data reporting”: How did you randomize your sites? Did you use GIS and place random points within some bounds of a polygon (e.g. yard or forest preserve)? Did you place a grid across the city and chose sites closes to a random intersect? Please be more specific.

Subsection “Citizen science camera trap surveys”: Please report the mean number of cameras deployed per volunteer (as an aside, please be consistent with the use of volunteer and citizen scientist).

Subsection “Citizen science camera trap surveys”: 200 m seems close together for mammal occupancy. Please justify the independence between sites, or, consider changing your terminology from true occupancy and maybe use habitat use instead.

Subsection “Citizen science camera trap surveys”: Were cameras rotated continuously throughout the year or did sampling only occur during a particular season. Please clarify? If they were rotated throughout the year, did you revisit sites or are some sites sampled in the earlier part of the calendar year being compared to sites sample later (let’s say early spring vs. late fall). If this is the case I would be worried that the sampling periods would violate the assumption of closure for your occupancy models. Please clarify and or address the issue.

Throughout the statistical analyses: It is clear that multiple authors wrote the different sections. Please take the time to be consistent in your tone, terminology, and methods descriptions throughout. For, example R is cited 3 different ways.

Subsection “Model covariates”: Please justify why you decided to use a single season occupancy model for multiple seasons with a year covariate instead of a dynamic occupancy model. The way you did it is not wrong but does not consider the dynamics between years. If you are violating closer (see comment above), you could sub-divide your data into appropriate seasons (considering closure) and use a multi-season occupancy model instead.

Subsection “Model covariates”: Please specify if you used NLCD land use data set and used an open space category or used the NLCD canopy cover dataset.

Subsection “Model covariates”: I am a bit confused here. So, you used the percentage of forested area within the 5 km buffer that was at least connected to a large continuous forest patch larger than 1km? Could that linkage continue outside the patch? For example, would a small 5m^2^ patch on the edge of the buffer be counted if it were connected to a larger forest patch that continues outside the buffer? It’s just a bit confusing, please clarify.

Subsection “Model covariates”: Also confused here. So now you measured the percentage of forest cover within a 100m radius? But it didn't have to be connected to continuous forest cover? So, a small vacant lot with some trees would be counted?

Subsection “Model covariates”: I have concerns about using number of detections/trap night as a metric of prey abundance (even just relative abundance). In our system, we get rabbits just sitting in front of the camera all day. How can you tease apart that a single rabbit didn't sit in front of the camera continuously triggering it at one site, but not another. For example, what if in a single night one rabbit sat in front of a camera for 100 minutes (100 detections) at site A and 100 rabbits passed in front of the camera a single time (100 detections) at site B? Very different abundance, but you get the same answer.

Subsection “Model covariates”: Does NDVI get at understory? Will a cell with really dense canopy cover but no understory give a different value than one with really dense canopy cover and thick understory? From what I have experienced, dense canopy cover in an urban park (with no understory) will often show up the same as a forested area if there are few gaps in the tree.

Subsection “Model covariates”: Did you use monthly NDVI, did you pick a month with peak greenness, or did you average across the month? I am assuming you used one value since you did not use a dynamic occupancy model, but please clarify.

Subsection “Model covariates”: Indicate which camera model was the reference.

Subsection “Detection rate models”: Did you have adequate fit for both your Poisson and occupancy models? Figure 4—figure supplement 1 has huge error bars, which could be a result of over parameterizing the model. Maybe report results from PPC in supplemental table. However, PPC checks for occupancy models are tough as they have trouble accounting for the correcting of detection. Cross validation is probably best. See Hooten and Hobbs (2015).

Subsection “Occupancy models”: I think you need to mention that you ran an occupancy model for a subset of species (and mention species) somewhere here. Unless you ran a multi-species model for all the species, in which case I don't see any mention of the rest of the species in the Results section.

Subsection “Occupancy models”: I am assuming that the city indicator is a 1 or a 0 based on your table in supplemental material, but this is not clear in the text. Please clarify how that interaction is formulated. You also need to report which city is represented by 1 and which city is represented by 0. Without this your tables don't mean much. You also need to report the intercept value of your model in your supplemental tables, so the reader can interpret results for the reference city.

Subsection “Comparison with global occupancy data”: Here is where the global comparison is first mentioned. I think you need a justification and a bit of a background in the introduction. Until I got to the Discussion section and saw the results, I was left wondering what the point of this analysis was.

Subsection “Study sites”: This passage is written a bit unclearly. As is, it sounds like you deliberately avoided urban areas in both cities. I suggest rephrasing for clarity.

Subsection “Model covariates”: I am assuming 'hunting' means hunting was allowed or hunting was recorded during the sample period? Be more specific.

Subsection “Model covariates”: I suggest putting the covariate abbreviation that you use in your models and tables in parentheses after each time they are mentioned in the text. I think this will help the reader follow along.

Subsection “Model covariates”: I suggest rephrasing to "We included an indicator (0 or 1) to categorize whether a site allowed hunting or not." Also, I assume that 0 indicates no hunting? But please be explicit in the text.

Subsection “Detection rate models”: Change 'count' to Poisson.

Subsection “Detection rate models”: I think you should say what your thinning rate was instead of just ith.

Subsection “Occupancy models”: What about plot type and all the other covariates?

---

## [Author Response]

Summary:The value of this paper is that it addresses a question of broad relevance with novel methods. The question is whether human population density near two US cities reduces the occurrence of large mammals in four types of habitats, large forests, forest fragments, open areas and yards. The assessment method involved 557 citizens operating cameras deployed for around 20 days at each of 1427 locations throughout a period of 4 years. In addition, the authors compared the results of this assessment with a global data set.The result was that mammal occurrence did not decline with human population density and even showed an opposite tendency, except for bobcats. Even though this results is not "strong" in the sense that the main point is the absence of a negative influence of human population density, it does send a very important message, in particular because it is based on such a large sampling scheme with a clear design of five human population density levels (wild < rural < exurban < suburban < urban) factorially crossed with the four habitat types (obviously, these two factors were not fully orthogonal because for example open areas and yards could not be found in the wild).Essential revisions:The interpretation of human population density as equivalent to disturbance is less important than implied by the authors and should be better justified. In particular, the comparison with the so-called intermediate disturbance hypothesis at the start of the Results and Discussion section seems unnecessary and unjustified, because it is not really tested later on nor even discussed. In fact, the curves drawn in Figure 1 are not really hump-shaped and probably do not have significant downwards curvature at high human population density (interpreted as high disturbance). The hump is even more elusive if Figure 3 is inspected.

We have removed the mention of IDH.

Currently, the discussion is somewhat US-centered. Although the two case-study cities are from that region, it would make the paper more relevant if comparisons with literature from other regions, including Europe, and involving other taxonomic groups would be included. In this context it would also be important to put the levels of human population densities in the two studied cities in relation to cities in the old world, which often occupy much smaller areas.

We have attempted to make comparisons with other cities around the world and added one example of other taxa (Results and Discussion section): “Our discovery of a wild suburbia suggests high levels of adaptation by mammals to developed landscapes over the last few decades, including predators and prey. […] Further understanding of how urban wildlife navigates human-dominated areas and factors that contribute to the adaptation of species to the Anthropocene will be critical to maintaining diversity in a rapidly urbanizing world.”

We hesitate to add too much on other taxa to the discussion for the sake of brevity and because the data we collected are relevant to mammals. The body of literature on other taxa (in particular birds and plants) is rich and we feel that further detailed discussion of those studies is not warranted here. However, if the editor feels strongly we will add a more in-depth discussion to that point.

The major issue you need to address in the revision is the description of methods. The detailed comments from the reviewers in this regard are listed below:I am not sure whether the results are influenced by a measurement bias, i.e. species tending to have higher activity in more disturbed regions and hence being more frequently recorded.

We would argue that higher activity indicates higher use of a particular habitat which is exactly what we are trying to show. We make no claims to density or absolute abundance, which would indeed be tricky due to differences in activity between gradient levels. What we are measuring (with occupancy and relative abundance) is essentially relative use between gradient levels, arguably a habitat preference.

Subsection “Detection rate models”: Either refer from here to the last paragraph of the “Occupancy models” subsection or mention already here how large (long) the burn-in phase was and report the value of i.

We have added this information and simplified the presentation “We based inference on posterior samples generated from three Markov chains, using trace plots to determine an adequate burn-in phase. All models achieved adequate convergence (R^≤1.1) (Gelman et al., 2014) by running for 50,000 iterations following a burn-in phase of 1000 iterations, thinning every 10 iterations.”

Subsection “Occupancy models”: To me it is not clear which species were paired: all ones, or just predators?

We have clarified this “We used the multispecies occupancy model of Rota et al., (2016), a generalization of the single-season occupancy model (MacKenzie et al., 2002) to accommodate two or more interacting species, to model the occupancy of four predator species: bobcat, coyote, red fox and gray fox.”

Subsection “Occupancy models”: Explain the elements of the formula (including subscripts).

We have added “Whereψ11 is the probability that both species occupy a site, ψ10 is the probability that only species 1 occupies a site,ψ01 is the probability that only species 2 occupies a site and ψ00 is the probability that neither species occupies a site.”

Subsection “Comparison with global occupancy data”: Be explicit on how this was assessed.

We have added clarification to this point “We summarized occupancy estimates of Rich et al. and our own study within each developmental level using a box and whisker plot and assessed statistically significant differences based on whether or not interquartile ranges overlapped.”

Currently, the information about the study design is placed in different parts of the manuscript so that it is difficult to get the overview. Supplementary file 1 is the most useful in this context, even though it would be easier if number of camera locations would also be shown (in addition to number of nights). It is not clear, if there was an additional spatial stratification that was not used in the analyses. For example, I could imagine that within each cell in Supplementary file 1 there were some spatial units and within these spatial units camera locations. Even if that was not the case, the question remains if spatial distance and arrangement of camera locations should have been considered.

Spatial replicates are indeed included in Supplementary file 1, in parentheses next to the camera nights. We also set a minimum spatial distance as part of our design (subsection “Citizen science camera trap surveys”) “All adjacent cameras were spaced at least 200m apart”.

Why did you use the crude models of rarefaction rather than obtain the inferences on species richness (including species accumulation) from the multi-species occupancy model of Rota et al.? See Dorazio et al. (2006) for how this can be done. Also, please cite the important work of Dorazio and Royle (2005) and Dorazio et al. (2006) as a the foundation on which models such as that by Rota et al. are built.

We suspect that the term “multispecies model” is a source of confusion. The “multispecies” model of Rota et al. is not a community model in the sense of Dorazio et al. (2006) and Dorazio and Royle (2005), but rather an extension of the single-species, single-season occupancy models of Mackenzie et al. (2002) and co-occurrence models of Richmond et al. 2010, with no community-level attribute estimators or focus, no species-level effects. We have clarified this in the Materials and methods section: “We used the multispecies occupancy model of Rota et al. (2016) to model the occupancy of four predator species: bobcat, coyote, red fox and gray fox. This model is distinct from the classic multispecies community models (Dorazio and Royle, 2005; Dorazio et al., 2006; Gelfand et al., 2005) and is rather a generalization of the single-season occupancy model (MacKenzie et al., 2002) to accommodate two or more interacting species.”

While we could formulate the model in a similar way to Dorazio/Royle community models, adding hyperparameters for richness, we think the rarefaction estimates are adequate for the story we are telling.

- Subsection “Model covariates” the sentence “We represented whether a site allowed hunting or not using 0/1.” sounds strange, rewrite.

We have changed this to “We included an indicator (0/1, no hunting/hunting) to categorize whether a site allowed hunting or not.”

- Subsection “Model covariates”: Please explain what a restrictive prior is.

We have added clarification: “We diagnosed univariate correlations between covariates using a Pearson correlation matrix, and used a restrictive prior for beta coefficients where correlation was >0.60 (i.e. logistic(0,1); a prior with reduced variance to induce shrinkage, similar to ridge regression; Hooten and Hobbs, 2015). All covariates were mean-centered.”

- Subsection “Occupancy models”: Really, the other basic groundstone on which your work is based are the multispecies occupancy models of Dorazio and Royle and of Gelfand (independently developed in 2005). These should also be cited.

We suspect that the term “multispecies model” is a source of confusion. The “multispecies” model of Rota et al., is not a community model in the sense of Dorazio and Royle and Gelfand, but rather an extension of the single-species, single-season occupancy models of Mackenzie et al., (2002) and co-occurrence models of Richmond et al., 2010, with no community-level attribute estimators or focus, no species-level effects. We have clarified this in the Materials and methods section: “We used the multispecies occupancy model of Rota et al. (2016) to model the occupancy of four predator species: bobcat, coyote, red fox and gray fox. This model is distinct from the classic multispecies community models (Dorazio and Royle, 2005; Dorazio et al., 2006; Gelfand et al., 2005) and is rather a generalization of the single-season occupancy model (MacKenzie et al., 2002) to accommodate two or more interacting species.”

*- Subsection “Occupancy models”:* “*We assumed all species occurred independently…”. Does this fit with what was just said “It contains single-species (first order)…”?*

We suspect this again is a product of some confusion over the model and the fact that it operates differently than the classic multispecies community models.

Subsection “Data reporting”: The first part of this paragraph is a bit weird to me. If you have to have a formal paragraph like this for the journal specifications, please ignore. As of now it reads like you are trying to convince the reader why an observational study was chosen instead of a true experiment. I think it is given that a camera trap study is going to be more of an observational study – we just try to standardize our sampling design as much as possible. If this paragraph is not required, I would just go straight into your methods – i.e. study design, site selection, sample size, etc.

While much of this information is required by the journal, we were able to pare it down and move it to other sections of the Materials and methods section, thus eliminating this first paragraph as the reviewer suggested.

Subsection “Data reporting”: How did you randomize your sites? Did you use GIS and place random points within some bounds of a polygon (e.g. yard or forest preserve)? Did you place a grid across the city and chose sites closes to a random intersect? Please be more specific.

We have clarified this: “Camera placement was randomized as much as possible using ArcMap (Version 10.1) to randomly generate points within polygons while following certain rules. For example, we selected sites within forests that volunteers were permitted to access and were within a reasonable hiking distance (i.e. < 11km hike round trip) with terrain that was not too steep to traverse safely (i.e. <45 degree slope). Within yards, cameras were placed as randomly as possible while avoiding the highest human traffic areas (i.e. walkways, doors, gates and driveways).”

Subsection “Citizen science camera trap surveys”: Please report the mean number of cameras deployed per volunteer (as an aside, please be consistent with the use of volunteer and citizen scientist).

We have clarified the instances of “citizen scientists” to “citizen scientist volunteers” to maintain consistency but also bring attention to the nature of those volunteers in a couple of key places. We have added: “Each individual camera was considered a “camera site” and volunteers ran cameras at an average of 2 sites each.”

Subsection “Citizen science camera trap surveys”: 200 m seems close together for mammal occupancy. Please justify the independence between sites, or, consider changing your terminology from true occupancy and maybe use habitat use instead.

We agree with the reviewer that this is not classical occupancy, in fact, all camera trap based studies violate the closure and independence assumptions since they are merely measuring the passage of animals in front, and not the longer term occupancy of a patch (Burton et al., 2015; Efford and Dawson, 2012). We have added: “Although we are using the term occupancy, because data were collected from camera traps estimates are more analogous to “use” than true occupancy (Burton et al., 2015).” We think this is sufficient to address the point and have left the terminology throughout as “occupancy”. We think this is a more established term than “use” and thus less confusing, however if the reviewer feels strongly we can change the terminology throughout.

Subsection “Citizen science camera trap surveys”: Were cameras rotated continuously throughout the year or did sampling only occur during a particular season. Please clarify? If they were rotated throughout the year, did you revisit sites or are some sites sampled in the earlier part of the calendar year being compared to sites sample later (let’s say early spring vs. late fall). If this is the case I would be worried that the sampling periods would violate the assumption of closure for your occupancy models. Please clarify and or address the issue.

We never revisited sites. Some sites are from the earlier part of the year, some from later with at least some samples within each season but not necessarily balanced. We have clarified this: “Cameras were deployed for three weeks and then moved to a new location without returning, with sampling taking place continuously throughout the year.”

Unfortunately, as is the nature of camera trapping, closure has already been violated and we have tried to clarify that in the Materials and methods section: “Although we are using the term occupancy, because data were collected from camera traps estimates are more analogous to “use” than true occupancy (Burton et al., 2015).”

With respect to temporal closure, it seems like we would expect occupancy estimates to be biased high as a result, but since our interest is more in relative occupancy (or use in this case) along the gradient, we believe that bias has little effect on our conclusions.

Throughout the statistical analyses: It is clear that multiple authors wrote the different sections. Please take the time to be consistent in your tone, terminology, and methods descriptions throughout. For, example R is cited 3 different ways.

We have made an effort to unify the tone and terminology throughout the methods.

Subsection “Model covariates”: Please justify why you decided to use a single season occupancy model for multiple seasons with a year covariate instead of a dynamic occupancy model. The way you did it is not wrong but does not consider the dynamics between years. If you are violating closer (see comment above), you could sub-divide your data into appropriate seasons (considering closure) and use a multi-season occupancy model instead.

Since we were covering such a large area and spatial variation in housing density was central to our question, we were most interested in gaining spatial replicates and less interested in temporal patterns, thus our design was such that cameras were not left in one place more than a month and therefore, to our knowledge, a dynamic occupancy model was not appropriate for this analysis.

Subsection “Model covariates”: Please specify if you used NLCD land use data set and used an open space category or used the NLCD canopy cover dataset.

We have clarified this “We used the Landscape Fragmentation Tool v2.0 (Vogt et al., 2007) and the NLCD (2006) land use dataset in ArcMap (Version 10.1) to create a landcover layer representing the percent of large core forest (forest patches larger than 1km^2^) in a 5km radius around camera locations.”

Subsection “Model covariates”: I am a bit confused here. So, you used the percentage of forested area within the 5 km buffer that was at least connected to a large continuous forest patch larger than 1km? Could that linkage continue outside the patch? For example, would a small 5m^2^ patch on the edge of the buffer be counted if it were connected to a larger forest patch that continues outside the buffer? It’s just a bit confusing, please clarify.

We have clarified this: “Forest patches did not necessarily fall entirely within the buffer.”

Subsection “Model covariates”: Also confused here. So now you measured the percentage of forest cover within a 100m radius? But it didn't have to be connected to continuous forest cover? So, a small vacant lot with some trees would be counted?

Yes indeed, the reviewer is correct that our intention was to take into account the small patches of habitat present in the surburban matrix, at fairly fine scales, which we surmised would be particularly important for shy species to navigate the urban/suburban zones. We added that clarification as well: “We used a 100m radius for small scale forest cover to best represent small forest patches within suburban neighborhoods (e.g. small vacant lots with trees, greenways).”

Subsection “Model covariates”: I have concerns about using number of detections/trap night as a metric of prey abundance (even just relative abundance). In our system, we get rabbits just sitting in front of the camera all day. How can you tease apart that a single rabbit didn't sit in front of the camera continuously triggering it at one site, but not another. For example, what if in a single night one rabbit sat in front of a camera for 100 minutes (100 detections) at site A and 100 rabbits passed in front of the camera a single time (100 detections) at site B? Very different abundance, but you get the same answer.

We agree with the reviewer that use of detections/day can be a problematic index of relative abundance, especially where individual photos are considered. To deal with this we considered only photographs separated by more than 1 minute as temporally independent records (see Parsons et al., 2016, for an analysis of the adequacy of this 1 minute interval for independence). Thus, if a rabbit sat in front of the camera for 100 minutes, it would still be counted as 1 rabbit, not 100. If 100 rabbits passed in front of the camera, assuming they did not pass in continuous succession without periods of time between of at least 1 minute, they would be counted as 100 rabbits. We have included this language in the Materials and methods section: “We grouped consecutive photos into on sequence if they were <60 seconds apart, and used these sequences as independent records, counting animals in the sequence, not individual photos (Parsons et al., 2016).”

Subsection “Model covariates”: Does NDVI get at understory? Will a cell with really dense canopy cover but no understory give a different value than one with really dense canopy cover and thick understory? From what I have experienced, dense canopy cover in an urban park (with no understory) will often show up the same as a forested area if there are few gaps in the tree.

As the reviewer points out, NDVI will not always reliably reflect understory, especially where canopy cover is dense. However, we felt that this was the best remote-sensing measurement available to try and account for this, however we also measured detection distance on the ground which accounts for thick undergrowth as well as the unique topography of each site which influenced detection probability. We have clarified this in the Materials and methods section: “To complement NDVI, we also considered site-specific detection distance, a measure of how far away the camera was able to detect a human, which is influenced by both understory and site topography.”

Subsection “Model covariates”: Did you use monthly NDVI, did you pick a month with peak greenness, or did you average across the month? I am assuming you used one value since you did not use a dynamic occupancy model, but please clarify.

We have added some clarification “Because both ambient temperature and undergrowth can affect the camera’s ability to detect an animal, we included daily covariates for temperature and NDVI (Moderate Resolution Imaging Land Terra Vegetation Indices 1km monthly, an average value over the month(s) the camera ran) obtained from Env-DATA (Dodge et al., 2013).”

Subsection “Model covariates”: Indicate which camera model was the reference.

We have added this clarification: “In Raleigh, two different camera models were used (both Reconyx and Bushnell) so we added a 0/1 (Bushnell/Reconyx) covariate to account for potential difference in detection probability between the two brands.”

Subsection “Detection rate models”: Did you have adequate fit for both your Poisson and occupancy models? Figure 4—figure supplement 1 has huge error bars, which could be a result of over parameterizing the model. Maybe report results from PPC in supplemental table. However, PPC checks for occupancy models are tough as they have trouble accounting for the correcting of detection. Cross validation is probably best. See Hooten and Hobbs (2015).

We have added the results of goodness-of-fit tests as a supplement (Supplementary file 3), using PPC for both. Since current methods of PPC with occupancy models have difficulty handling detection, as the reviewer points out, we derived a novel (to our knowledge) method for posterior predictive checks of the occupancy model and have included that proof as Supplementary file 4. We have added this to the Materials and methods section: “We assessed model fit with posterior predictive checks (PPC) (Gelman et al., 2014; Kery and Schaub, 2012) by calculating the sum of squared Pearson residuals from observed data (*T(y*)) and from data simulated assuming the fully parameterized model was the data-generating model (*T(y_sim_*)). We calculated a Bayesian *p*-value as *p_B_* = Pr(*T(y_sim_*) > *T(y*)) from posterior simulations and assumed adequate fit if 0.1 < *p_B_* < 0.9. To our knowledge, the squared Pearson’s residual has not been derived in the context of occupancy models, so we present our derivation of this test statistic in Supplementary file 4. We added a random effect on detection/non-detection for the coyote portion of the model since initial assessments of fit for this species were inadequate (i.e. p_B_>0.9).”

Subsection “Detection rate models”: “Both of our models fit well, with Bayesian p-values between 0.1 and 0.9 (Supplementary file 3).” The additional random effect placed on coyote detection/non-detection to improve fit did not substantially change estimates of psi or associated uncertainty.

Subsection “Occupancy models”: I think you need to mention that you ran an occupancy model for a subset of species (and mention species) somewhere here. Unless you ran a multi-species model for all the species, in which case I don't see any mention of the rest of the species in the Results section.

We have added: “We used the multispecies occupancy model of Rota et al., (2016) to estimate the probability of occupancy of four predator species: bobcat, coyote, red fox and gray fox.”

Subsection “Occupancy models”: I am assuming that the city indicator is a 1 or a 0 based on your table in supplemental material, but this is not clear in the text. Please clarify how that interaction is formulated. You also need to report which city is represented by 1 and which city is represented by 0. Without this your tables don't mean much. You also need to report the intercept value of your model in your supplemental tables, so the reader can interpret results for the reference city.

We have added that clarification: “We considered interactions (i.e. city*covariate) between each occupancy covariate and city (0/1, DC/Raleigh).”

We have also added the intercepts into Supplementary file 5, where they were missing.

Subsection “Comparison with global occupancy data”: Here is where the global comparison is first mentioned. I think you need a justification and a bit of a background in the introduction. Until I got to the Discussion section and saw the results, I was left wondering what the point of this analysis was.

We have attempted to add some more clarity of structure and purpose to the end of the Introduction, including about our comparison with global wildlands: “Our objectives were to determine how diversity, richness, detection rate, and occupancy of the mammal community changes as a function of human disturbance. We hypothesized that the availability of supplemental food at higher levels of development would positively affect mammalian populations and outweigh the negative effects of disturbance, except for the most sensitive species. Specifically, we predicted that relative abundance and occupancy of mammals would increase with developmental level but that species richness and diversity at these higher levels would be lower. Furthermore, we predicted that the occupancy of the most sensitive mammal species (i.e. large and medium carnivores) would be lower at the highest development levels compared to wild areas both in our study area and around the world.”

Subsection “Study sites”: This passage is written a bit unclearly. As is, it sounds like you deliberately avoided urban areas in both cities. I suggest rephrasing for clarity.

We have rephrased to clarify: “Washington, District of Columbia, USA (hereafter DC) is a city of approximately 177km^2^ with an estimated human population size of 681,000, thus a density of 3,847 people/km^2^. Our study spanned a 56,023.7km^2^ area around the city with a mean of 4.4 houses/km^2^ and matrix of agriculture (~21.3%) and forest (~54.1%). Raleigh, North Carolina, USA (hereafter Raleigh) is approximately 375km^2^ with an estimated human population size of 459,000, thus a density of 1,278 people/km^2^. Our study spanned a 66,640km^2^ area around the city with a mean of 17.7 houses/km^2^ and matrix of agriculture (~24.3%) and forest (~52.3%).”

Subsection “Model covariates”: I am assuming 'hunting' means hunting was allowed or hunting was recorded during the sample period? Be more specific.

We have added this clarification: “We modeled variation in occupancy (ψ) using 13 covariates (Supplementary file 2) representing development level, the amount of core forest, small scale forest cover, prey relative abundance and whether hunting was allowed.”

Subsection “Model covariates”: I suggest putting the covariate abbreviation that you use in your models and tables in parentheses after each time they are mentioned in the text. I think this will help the reader follow along.

While we agree with the reviewer in principle, we also think that this will clutter the text significantly and may reduce readability, so we have chosen not to include the abbreviations in the text.

Subsection “Model covariates”: I suggest rephrasing to "We included an indicator (0 or 1) to categorize whether a site allowed hunting or not." Also, I assume that 0 indicates no hunting? But please be explicit in the text.

We have made the suggested change and added some clarifications about the reference level: “We included an indicator (0/1, no hunting/hunting) to categorize whether a site allowed hunting or not.”

Subsection “Detection rate models”: Change 'count' to Poisson.

We feel this is not necessary since we mention above that it is a Poisson distributed count model and calling it a “count” model is statistically correct and easy to understand. If the reviewer feels strongly, however, we are willing to change it.

Subsection “Detection rate models”: I think you should say what your thinning rate was instead of just ith.

We have added more clarification: “All models achieved adequate convergence (R^≤1.1) (Gelman et al., 2014) by running for 50,000 iterations following a burn-in phase of 1000 iterations, thinning every 10 iterations.”

Subsection “Occupancy models”: What about plot type and all the other covariates?

We have added: “We based predictor significance on whether beta coefficient 95% credible intervals overlapped zero.”

We did not assess differences in occupancy between plot types, in part because detections were so split (i.e. basically no bobcats or coyotes in yards so difficulty with occupancy model convergence). Therefore, we felt this trend would be more easily illustrated with the count model.